# Cooperative Multi-Agent Reinforcement Learning with Sequential Credit Assignment

## Abstract

Centralized training with decentralized execution is a standard paradigm for cooperative multi-agent reinforcement learning (MARL), with credit assignment being a major challenge. In this paper, we propose a cooperative MARL method with sequential credit assignment (SeCA) that deduces each agent's contribution to the team's success one by one to learn better cooperation. We first present a sequential MARL framework, under which we introduce a new counterfactual advantage to evaluate each agent based on its preceding agents' actions in a specific sequence. As this credit assignment sequence tremendously impacts the performance, we further present a sequence adjustment algorithm utilizing integrated gradients. It dynamically modifies the sequence among agents according to their contribution to the team. SeCA employs a network which either estimates the Q value for training the centralized critic or deduces the proposed advantage of each agent for decentralized policy learning. Our method is evaluated on a challenging set of StarCraft II micromanagement tasks and achieves state-of-the-art performance.

## 1 Introduction

Cooperative multi-agent reinforcement learning (MARL) is a helpful tool in numerous applications such as robot swarm control [9], autonomous vehicle coordination [3], network routing [36], and productivity optimization [37]. This kind of problem where agents learn coordinated policies to optimize the global reward has been extensively studied in recent years [7, 19, 18, 38, 8].

One natural way of addressing the cooperative MARL problem is the *centralized* approach, which treats the team as a single actor with a joint action space. Although we can trivially apply single-agent reinforcement learning algorithms to such settings, it usually does not scale well because the size of the joint action space grows exponentially with the number of agents. Besides, it is not applicable in real-world settings due to the inherent constraints on agent observability and communication. An alternative approach is to learn *decentralized* policies by independently training agents based on their local observations, but simultaneous exploration often brings non-stationarity that causes unstable learning and difficulties in convergence. As a result, the majority of work on MARL follows the *centralized training with decentralized execution* (CTDE) paradigm [17, 10, 22, 6], where decentralized policies can access extra state information during training.

A crucial challenge of the CTDE paradigm in cooperative settings is to correctly deduce each agent's contribution to the team's success, also known as the *multi-agent credit assignment problem* [4]. Existing methods can be classified as *implicit* and *explicit* credit assignment [39]. Previous implicit methods often deduce all agents' contributions by representing the global state-action value as an aggregation of each agent's state-action value [26, 22, 12, 24, 21, 29] and assigning the shared rewards to each agent according to the joint action at one time. In this way, these methods avoid the complex interaction analysis and instead fit these cooperation relationships by neural networks. However,

implicit methods often face limitations in expressiveness, and their extensions to continuous action spaces may require additional strategies [39].

On the other hand, recognized explicit approaches calculate difference rewards [34] against a certain reward baseline [28, 20, 6]. However, in cooperative MARL, evaluating any agent's action requires considering the actions of all agents, so it is often difficult to determine the impact a particular agent's behavior has on the team when we have not assessed other agents' actions. In other words, we can not say that a single agent's action is bad if the team receives a small reward because the shared reward is not decided only by this agent's behavior. Maybe its action is actually good in that state.

This paper presents a sequential credit assignment SeCA to evaluate individual agent actions explicitly and sequentially. Our motivation is to address the drawbacks of implicit methods that neglect the cooperation between agents or simply leave it to neural networks and further improve explicit credit assignment. In summary, we face two main challenges to learn a better explicit credit assignment: (1) how to alleviate the problem that it is hard to accurately deduce the contribution of one agent without previously assessing all the others' action, and (2) how to evaluate agents better in an explicit way.

To deal with (1), we introduce a sequential MARL framework. As mentioned above, without assessing the behaviors of other agents, we would never be able to evaluate a given agent's action accurately. However, we point out in this paper that some agents are less affected by such influences than others, and we can first assign credit to them. For instance, evaluating a staff's action needs to take the CEO's command or action into consideration, while the former has little importance in assessing the CEO. Thus, we could evaluate the CEO first without considering the staff's behavior and then analyze the staff based on the CEO's action. We fully consider the action coordination between agents and explicitly deduce contribution to them one by one according to a particular order, so as to make up for the disadvantage of implicit methods that the cooperation is only inexplicably fitted by neural networks. Intuitively, the order significantly impacts the overall performance, so we further propose an algorithm to adjust the sequence dynamically through integrated gradients [25].

As for (2), we compute an advantage function for each agent to attribute agent contributions explicitly. COMA [6] is a representative method that computes a baseline for each agent to reason about counterfactuals in which only one agent $a$'s action changes, so its evaluation of $a$'s action is based on the joint action $\mathbf{u}^{-a}$ of other agents. In other words, the policy gradient of COMA only encourages agent $a$ to learn in the direction that benefits the team while other agents are acting $\mathbf{u}^{-a}$, but the others' actions are not necessarily $\mathbf{u}^{-a}$ when executing. Unlike COMA, we focus more on the action coordination among agents and propose a new advantage under the proposed sequential framework.

We summarize the contributions of this paper as follows: (1) We propose a sequential MARL framework in Section 3.2; (2) Under this framework, we introduce a sequential advantage function for each agent to guide their learning explicitly in Section 3.3. We further prove that the sequential credit assignment we proposed achieves additive advantage-decomposition. (3) We present a sequence adjustment algorithm based on integrated gradients to modify the credit assignment order dynamically in Section 3.4. This algorithm alleviates the impact caused by the sequence's randomness and helps achieve competitive performance on a challenging set of StarCraft II micromanagement tasks [23].

## 2   Related Work

Explicit credit assignment gives valuable insights into agent actions' contributions to the shared team reward and substantially promotes policy optimization. The representative method COMA [6] utilizes a counterfactual baseline that marginalizes out a single agent's action while keeping the other agents' actions fixed to calculate the advantage function. However, the advantage evaluates a single agent's action based on the other agents' current behaviors and ignores different action combinations. SQDDPG [30] distributes the global reward reflecting each agent's contribution through Shapley Value. Although SQDDPG provides a theoretically justified framework, its assumption on the observability and convex game makes it impractical and performs poorly in complex environments.

Implicit methods are a more common way when addressing the credit assignment challenge. Among them, LICA [39] is a policy-based method, which learns an end-to-end differentiable optimization where it trains a hypernetwork that maps the state into a set of weights which, in turn, maps the action policies into the Q estimate. On the other hand, value-based methods often represent the global state-action value as an aggregation of the individual values. The value decomposition is linear

in the earlier work VDN [26], and it ignores the state information. QMIX [22] learns a non-linear mixing network with the global state and maps the individual state-action values into the joint Q value estimate. Although QMIX performs well in various environments, it still faces the mixing network's monotonicity constraint limitation. QTRAN [24] further avoids the representation limitations by using linear constraints between individual utilities and the global state-action value. It guarantees optimal decentralization, but its constraints are computationally intractable, and the relaxations often lead to unsatisfied performance. QPLEX [29] decomposes Q values following the dueling structure, transferring the monotonicity condition from Q values to advantage values. QPD [35] leverages the integrated gradient attribution technique to decompose global Q values along trajectory paths based on the assumption that an agent's local reward is linearly correlated with its contribution to the team.

## 3 Methods

### 3.1 Preliminaries

**Notations.** This work considers a fully cooperative multi-agent task with $n$ agents $\mathcal{A} = \{1, ..., n\}$ as a Dec-POMDP [16] defined by a tuple $G = (S, U, P, r, Z, O, n, \gamma)$. The environment has a true state $s \in S$. Each agent $a$ chooses an action $u_t^a$ from its action space $U$ at each timestep $t$ and forms a joint action $\mathbf{u}_t$ that induces a transition in the environment according to the state transition function $P(s_{t+1}|s_t, \mathbf{u}_t) : S \times U^n \times S \to [0, 1]$. The agents share the same reward function $r(s, \mathbf{u}) : S \times u^n \to \mathbb{R}$, and $\gamma \in [0, 1)$ is the discount factor. We consider partially observable scenarios in which agent $a$ acquires its local observation $z^a \in Z$ drawn from $O(s_t, a) : S \times \mathcal{A} \to Z$. Each agent has an action-observation history $\tau^a \in T \equiv (Z \times U)^*$, on which it conditions a policy $\pi^a(u^a|\tau^a) : T \times U \to [0, 1]$. We denote joint quantities over agents in bold and joint quantities over agents other than a given agent $a$ with the superscript $-a$.

**Integrated Gradients.** Many works aim to attribute the predictions of deep networks to their input features [1, 15, 2]. As one of them, integrated gradients [25] aggregates the gradients along the inputs that fall on the lines between the baseline $\vec{b}$ and the input $\vec{x} = (x_1, ..., x_j, ..., x_d)$. It explains how much one feature affects the deep network output $F$ while changing from $F(\vec{b})$ to $F(\vec{x})$ along a path between $\vec{b}$ and $\vec{x}$. Given a path function $\tau(\alpha)$ with $\alpha \in [0, 1]$ specifying a path from baseline $\tau(0) = \vec{b}$ to the input $\tau(1) = \vec{x}$, then integrated gradients along the $j^{th}$ dimension is acquired by:

$$c_j = \text{PathIG}_j^\tau(\vec{x}) ::= \int_0^1 \frac{\partial F(\tau(\alpha))}{\partial \tau_j(\alpha)} \frac{\partial \tau_j(\alpha)}{\partial \alpha} \, d\alpha, \tag{1}$$

where $c_j$ represents $x_j$'s contribution to the difference between baseline prediction $F(\vec{b})$ and $F(\vec{x})$. In this work, we leverage the integrated gradients technique to dynamically adjust the order of our proposed sequential credit assignment according to each agent's contribution to the team.

### 3.2 Sequential MARL Framework

The relationship in a multi-agent system is complicated, as every agent makes decisions based on the environment interfered with by the other agents. If we model each agent as a node and model the cooperations between them as edges, the cooperative relationship will be built as a complicated web-like graph shown in Figure 1(a). Evaluating the actions of any agent should take into account the behaviors of other agents in this situation. It is hard to judge whether an agent's current action is beneficial to the team when we have not evaluated other agents' actions. If we cannot determine an analysis order, we can only analyze all the agents implicitly as most existing methods did, and the cooperation is often fitted only by deep neural networks, leading to unsatisfactory results.

This section presents a sequential framework for cooperative MARL, which aims to analyze agents' actions one by one. Our key assumption is that evaluations of some agents in a team are less affected than others. Thus we can study these less-affected agents first and then analyze the others based on the actions of these already-studied agents. For instance, when evaluating a staff's action, the CEO's decision plays a vital role because we have to judge whether the staff obeys the command or not. On the contrary, the staff intuitively has little impact on evaluating the CEO's decision. In assessing the CEO, we often consider external factors such as market situation, modeled as state $s$ in MARL.

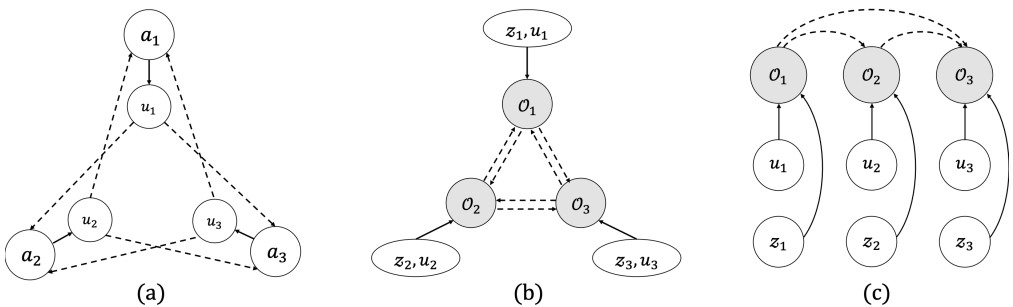

Figure 1: A toy example with three agents. (a) Agents affect each other as they choose actions based on the state interfered with by the others' actions. (b) The study on one agent will influence all the other agents' assessments in the original MARL framework. Agent's cooperation analyses are interrelated. (c) Each agent's cooperation study in the proposed sequential MARL framework. Dotted arrows representing correlations decrease from 6 in (b) to 3 in (c), reducing the complexity by half. This merit also holds for systems with other numbers of agents.

We introduce a variable $\mathcal{O}_i$ to help model this sequential MARL framework. This additional variable represents a random event that our cooperation study (e.g., credit assignment) on agent $a_i$ is optimal or precise. Then the probability $p(\mathcal{O}_i)$ denotes the accuracy of our research on agent $a_i$. For illustration and understanding convenience, we discuss a simple multi-agent system with three agents as an example, in which agents are identified by $a_i$ ($i \in \{1, 2, 3\}$). In original MARL, the evaluation of agent $a_i$ will influence all the other agents' assessments. Thus events $\mathcal{O}_1$, $\mathcal{O}_2$ and $\mathcal{O}_3$ are mutually dependent, as shown in Figure 1(b). We calculate the probability of studying the system accurately by computing conditional probabilities:

$$p(\mathcal{O}_1, \mathcal{O}_2, \mathcal{O}_3) = p(\mathcal{O}_1) \cdot p(\mathcal{O}_2|\mathcal{O}_1) \cdot p(\mathcal{O}_3|\mathcal{O}_1, \mathcal{O}_2) \tag{2a}$$

$$\vdots$$

$$= p(\mathcal{O}_3) \cdot p(\mathcal{O}_2|\mathcal{O}_3) \cdot p(\mathcal{O}_1|\mathcal{O}_2, \mathcal{O}_3) \tag{2b}$$

where $p(\mathcal{O}_j|\mathcal{O}_i)$ denotes the probability of agent $a_j$'s accurate analysis under the condition of conducting a precise study on agent $a_i$. It also indicates the accuracy of $a_j$'s analysis conditions on precisely assess $a_i$. We then conclude that:

$$p(\mathcal{O}_1, \mathcal{O}_2, \mathcal{O}_3) = p(\mathcal{O}_i) \cdot p(\mathcal{O}_j|\mathcal{O}_i) \cdot p(\mathcal{O}_k|\mathcal{O}_i, \mathcal{O}_j) \tag{3}$$

where $i, j, k \in \{1, 2, 3\}, i \neq j, k \neq i, j$.

We take Equ.(2a) as an example. To study the cooperation of this multi-agent system precisely (i.e., big $p(\mathcal{O}_1, \mathcal{O}_2, \mathcal{O}_3)$), we can first analyze $a_1$ as accurately as possible (i.e., big $p(\mathcal{O}_1)$) and then go on to investigate $a_2$ and $a_3$ respectively with the best possible accuracy (i.e., big $p(\mathcal{O}_2|\mathcal{O}_1)$ and $p(\mathcal{O}_3|\mathcal{O}_1, \mathcal{O}_2)$) under the condition of preceding agents' precise analysis.

The sequential MARL framework reduces the complexity of the model with six dotted arrows that indicate correlations between agents' evaluations in Figure 1(b) by half, as those three dotted lines in Figure 1(c) show. Equ.(3) suggests that we can analyze the cooperation of a multi-agent system in any order, but from the CEO-Staff example, we can see that the difficulty of analyzing in various orders is not the same. Further discussion on the sequence will show in Section 3.4.

In general, we specify an order to analyze the cooperation in the sequential MARL framework. We fix an agent's actions after assessing it and study a particular agent based on the fixed actions of its preceding agents, reflecting the intuition that a CEO's decision has a strong influence on evaluating the staff in the example mentioned earlier. This sequential MARL framework significantly alleviates the correlations in studying agents and helps us assess their cooperation more directly.

### 3.3 Sequential Credit Assignment

Following the CTDE paradigm, we utilize a centralized critic for each actor to follow a gradient based on an advantage function $A$ estimated from this critic:

$$g = \nabla_{\theta^\pi} \log \pi \left(u|\tau_t^a\right) A. \tag{4}$$

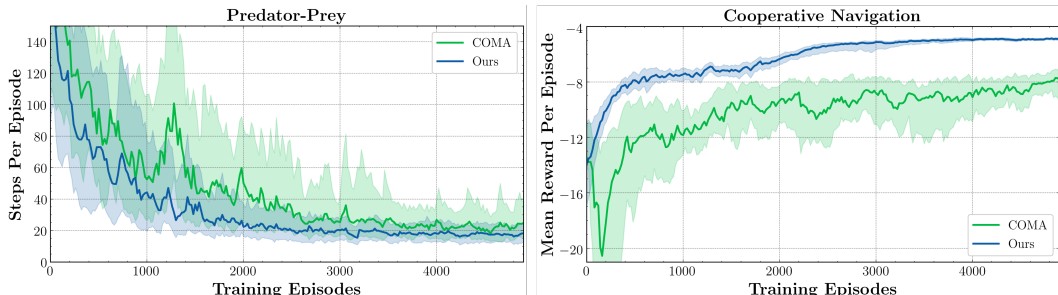

Figure 2: Performances between COMA's counterfactual advantage and ours in two environments. (Left) *Predator-Prey*. Three predators cooperate to chase a faster prey that acts randomly in an area containing two obstacles. The game terminates when a predator captures the prey, and then a shared reward is given. The predators trained by our advantage capture the prey faster. (Right) *Cooperative Navigation* initializes three agents and three landmarks with random locations. Agents cooperate to cover all the landmarks, and the shared reward is the negative sum of displacements between each landmark and its nearest agent. Our method helps the team gain bigger rewards than COMA.

The advantage function $A$ for each actor explicitly deduces how that particular agent contributes to the team. COMA [6] introduced a counterfactual baseline inspired by difference rewards [34]. For each agent $a$, COMA computes an advantage function that compares the Q-value for the action $u^a$ to a counterfactual baseline that marginalizes out $u^a$ while keeping the others' actions $\mathbf{u}^{-a}$ fixed:

$$A^a_{COMA}(s,\mathbf{u}) = Q\left(s,\left(u^a,\mathbf{u}^{-a}\right)\right) - \sum_{u'^a}\pi^a\left(u'^a|\tau^a\right)\cdot Q\left(s,\left(\mathbf{u}^{-a},u'^a\right)\right). \tag{5}$$

COMA avoids expensive calculations through careful network design. However, each agent's contribution deduced by COMA is still imperfect. The evaluation of $u^a$ is based on the fixed $\mathbf{u}^{-a}$ in Equ.(5), so agent $a$ will learn a policy that works better with $\mathbf{u}^{-a}$ in this way. It ignores the joint actions $(u^a,\mathbf{u}^{-a'})$ with $\mathbf{u}^{-a'} \neq \mathbf{u}^{-a}$ that may lead to unexpected results when assessing $u^a$.

To analyze each agent $a$'s contribution more objectively, we consider the influence of all joint actions with $u^a$. Considering all potential action combinations, we calculate a counterfactual advantage for each agent's action, derived by computing the expectation on all the actions of other agents:

$$A^a(s,\mathbf{u}) = \mathbb{E}_{\mathbf{u}^{-a}}\left[Q\left(s,\left(u^a,\mathbf{u}^{-a}\right)\right)\right] - \mathbb{E}_{\mathbf{u}^{-a}}\left[\sum_{u'^a}\pi^a\left(u'^a|\tau^a\right)\cdot Q\left(s,\left(\mathbf{u}^{-a},u'^a\right)\right)\right]. \tag{6}$$

Under our proposed sequential MARL framework, we carry out credit assignment according to a specific order, and there is no need to consider all the possible joint actions. After assessing agent $a$, we fix its action and evaluate agents after it based on $a$'s fixed action, so the following agents' credit assignments do not have to compute the expectation on $u^a$ anymore.

We now give the detailed sequential credit assignment for a team with $n$ agents identified by $a_i(i \in \{1,...,n\})$ under one specific sequence $\{a_1,a_2,...,a_n\}$, and it can also be concluded from the rest $(n!-1)$ orders in the same way. Here we denote $\mathbf{u}^{a_{i-1}}_{a_1} = [u^{a_1},u^{a_2},...,u^{a_{i-1}}]$ $(i=2,3,...,n)$.

As for agent $a_i$ $(i \neq 1)$ in the sequence, the contribution of its leading agents $a_1,a_2,...,a_{i-1}$ has been deduced. We fix the leading agents' actions and assess agent $a_i$'s action based on $\mathbf{u}^{a_{i-1}}_{a_1}$, so there is no need to calculate the expectations on $[u^{a_1},u^{a_2},...,u^{a_{i-1}}]$, simplifying Equ.(6) to:

$$A^{a_i}(s,\mathbf{u}) = \sum_{u'^{a_{i+1}}}\cdots\sum_{u'^{a_n}}\pi^{a_{i+1}}\left(u'^{a_{i+1}}|\tau^{a_{i+1}}\right)\cdots\pi^{a_n}\left(u'^{a_n}|\tau^{a_n}\right)\cdot Q\left(s,\left(\mathbf{u}^{a_i}_{a_1},u'^{a_{i+1}},\cdots,u'^{a_n}\right)\right)$$

$$- \sum_{u'^{a_i}}\cdots\sum_{u'^{a_n}}\pi^{a_i}(u'^{a_i}|\tau^{a_i})\cdots\pi^{a_n}\left(u'^{a_n}|\tau^{a_n}\right)\cdot Q\left(s,\left(\mathbf{u}^{a_{i-1}}_{a_1},u'^{a_i},\cdots,u'^{a_n}\right)\right). \tag{7}$$

Then the first agent $a_1$'s advantage is:

$$A^{a_1}(s,\mathbf{u}) = \sum_{u'^{a_2}}\cdots\sum_{u'^{a_n}}\pi^{a_2}\left(u'^{a_2}|\tau^{a_2}\right)\cdots\pi^{a_n}\left(u'^{a_n}|\tau^{a_n}\right)\cdot Q\left(s,\left(u^{a_1},u'^{a_2},\cdots,u'^{a_n}\right)\right)$$

$$- \sum_{u'^{a_1}}\cdots\sum_{u'^{a_n}}\pi^{a_1}\left(u'^{a_1}|\tau^{a_1}\right)\cdots\pi^{a_n}\left(u'^{a_n}|\tau^{a_n}\right)\cdot Q\left(s,\left(u'^{a_1},u'^{a_2},\cdots,u'^{a_n}\right)\right) \tag{8}$$

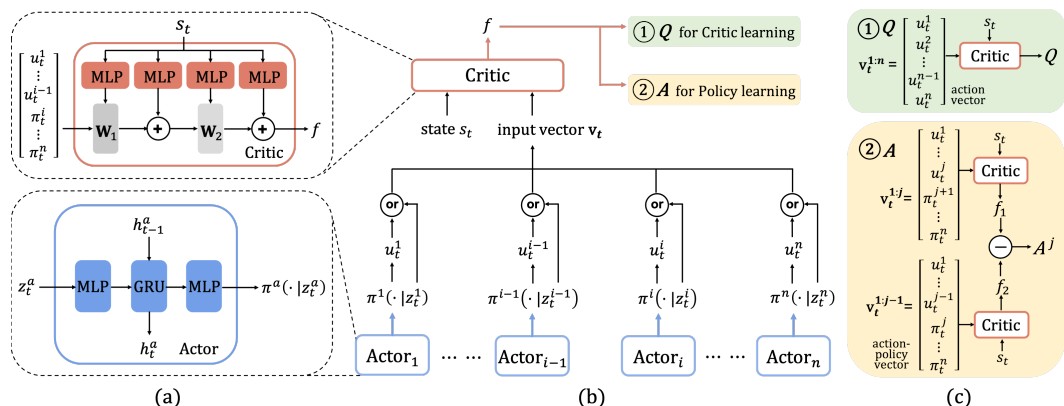

Figure 3: (a) A centralized mixing critic network that maps the state into a set of weights (top) and the decentralized agent network structure (bottom). (b) The overall SeCA architecture. (c) Critic learning (top) and policy learning (bottom) flow. View in color if possible for better understanding.

To illustrate the effectiveness of our sequential counterfactual advantage, we conduct a simple but illuminating test in two common multi-agent particle environments [11], *Predator-Prey* and *Cooperative Navigation*. We train both methods with 5 random seeds, and agents are trained for 5000 episodes. We provide detailed information on the environments and experiments in the Appendix. As shown in Figure 2, our sequential advantage functions help agents handle the task faster and better.

Our sequential advantage for each agent achieves an additive decomposition of the total advantage function, which to some extent explains the soundness and superiority of our advantage over COMA's.

**Claim 1.** *The proposed sequential credit assignment achieves additive advantage-decomposition.*

*Proof.* See Appendix A. ∎

Facing the same problem as COMA that those evaluations are expensive, we model the first term in Equ.(7) as a function $f_\phi$ of $(u^{a_1}, u^{a_2}, ..., u^{a_i}, \pi^{a_{i+1}}, ..., \pi^{a_n})$ to address this issue, and the second term is a similar function of $(u^{a_1}, u^{a_2}, ..., u^{a_{i-1}}, \pi^{a_i}, ..., \pi^{a_n})$. Thus, we rewrite Equ.(7) as:

$$A^{a_i} = f_\phi\left(s; u^{a_1}, u^{a_2}, ..., u^{a_i}, \pi^{a_{i+1}}, ..., \pi^{a_n}\right) - f_\phi\left(s; u^{a_1}, u^{a_2}, ..., u^{a_{i-1}}, \pi^{a_i}, ..., \pi^{a_n}\right). \quad (9)$$

Here $f_\phi$ is a function evaluating agents' action-policy vectors, where $f_\phi\left(u^{a_1}, u^{a_2}, ..., u^{a_n}\right) = Q$ and $f_\phi\left(\pi^{a_1}, \pi^{a_2}, ..., \pi^{a_n}\right) = V$. We design the complete setup for SeCA, which is illustrated in Figure 3.

**Critic Learning.** We train critic $f_\phi$ on-policy to estimate $Q$, utilizing a practical variant of TD($\lambda$) [27] adapted for use with deep neural networks. In particular, the critic parameter $\phi$ is updated by minibatch gradient descent to minimize the following loss:

$$\mathcal{L}_t(\phi) = \left(y_t^{(\lambda)} - f_\phi(s_t, \mathbf{u}_t)\right)^2, \text{ where } y_t^{(\lambda)} = r_t + \gamma\left(\lambda y_{t+1}^{(\lambda)} + (1-\lambda)f_{\phi^-}\left(s_{t+1}, \mathbf{u}_{t+1}\right)\right). \quad (10)$$

We utilize a target critic $f_{\phi^-}$ [14] to improve learning stability and update $\phi^- \leftarrow \phi$ periodically. The critic learning flow is shown at the top of Figure 3(c). The input for critic training is the state $s$ and the action vector $\mathbf{u} = \left[u^1, u^2, ..., u^n\right]$ denoted as $\mathbf{v}^{1:n}$.

**Policy Learning.** We optimize each agent $a$'s policy parameter $\theta_a$ by maximizing the following objective, which contains our proposed advantage function and an entropy regularization term $\mathcal{H}$:

$$g^a = \mathbb{E}_{\tau \sim \pi}\left[\nabla_{\theta_a} \log \pi^a(u^a|\tau^a)A^a(s, \mathbf{u}) + \mathcal{H}\left(\pi^a(\cdot|\tau^a)\right)\right], \quad (11)$$

where the derivative of the adaptive entropy regularization term $\mathcal{H}(\pi^a(\cdot|\tau^a))$ [39] with respect to the $i$-th action probability $p_i^a$ is given by:

$$d\mathcal{H}_i := -\xi \cdot (\log p_i^a + 1)/H(\pi^a(\cdot|\tau^a)), \text{ where } H(\pi^a(\cdot|\tau^a)) = \mathbb{E}_{u^a \sim \pi^a}\left[-\log \pi^a(u^a|\tau^a)\right]. \quad (12)$$

We share parameters among agents, and the gradient we use to train the actor shared by all agents is:

$$g = \mathbb{E}_{\tau \sim \pi}\left[\sum_a \left(\nabla_{\theta_a} \log \pi^a(u^a|\tau^a)A^a(s, \mathbf{u}) + \mathcal{H}\left(\pi^a(\cdot|\tau^a)\right)\right)\right]. \quad (13)$$

The inputs of the centralized critic $f_\phi$ to compute the advantage function are the state $s$ and two action-policy vectors $\mathbf{v}^{1:i} = \left[u^1, ..., u^i, \pi^{i+1}, ..., \pi^n\right]$ and $\mathbf{v}^{1:i-1} = \left[u^1, ..., u^{i-1}, \pi^i, ..., \pi^n\right]$. The bottom of Figure 3(c) demonstrates the policy learning flow.

## 3.4 Sequence Adjustment Through Integrated Gradients

We apply integrated gradients to adjust the credit assignment sequence dynamically. Reviewing the enlightening and straightforward CEO-Staff example discussed in Section 3.2, we can evaluate the staff's behavior based on the CEO's decision, but assessing the CEO does not require much attention to the staff's action. Therefore, we would analyze the CEO first and then evaluate the staff based on the CEO's current action. However, this example is not generalized for two reasons: (1) There are often multiple agents taking the same role in a system with superior-subordinate relationships, and the sequence of these agents is hard to determine; (2) Not all scenarios have such superior-subordinate relationships. The agents often do not need to follow others' commands in many applications.

We generalize the CEO-Staff example to propose a universal model. Instead of focusing on the roles among the agents as in [31, 32], we are more interested in agents' contributions. Although the CEO and the staff have a superior-subordinate relationship, they are essentially employees of an enterprise. The staff plays an auxiliary role and acts based on the CEO's decision. The staff's work is meaningful only if the CEO's decision is correct. Therefore, we often intuitively assume that an enterprise's leader is paid more and contributes more. Based on this, we transform the roles of the CEO and staff into employees with different contributions to the enterprise. In the sequential MARL framework, we first assign credit to the agent with a higher contribution to the team.

The attribution method is a powerful way to determine the influence of input features' each component on the network output value [2]. Among them, integrated gradients [25] leverages path integral to aggregate gradients along the inputs that fall on the lines between the baseline and the input, which is a natural tool for measuring each agent's contribution. QPD [35] utilizes the integrated gradient attribution technique to decompose shared rewards along trajectory paths, revealing how much each agent's observation and action contributes to the global Q value. However, it remains unclear whether individual Q value should be linearly correlated to or approximated by the agent's contribution, as in the case of QPD. The proper connection between agents' contributions and their individual Q values in a cooperative team is worth well studied for the community.

Here we avoid detailed analysis on the relationship between agents' contributions and their individual rewards. Instead, we use integrated gradients to measure agents' contributions to the state transition and adjust the credit assignment sequence based on their contributions. In particular, we estimate agent $a$'s contribution $c^a$ in the trajectory path $\tau_{t_1}^{t_2}$ from time $t_1$ to $t_2$ based on its policy vector $\pi^a$:

$$c^a = \sum_{x_j \in \pi^a} \text{PathIG}_j^{\tau_{t_1}^{t_2}}(\pi^a), \tag{14}$$

where $x_j$ is $j$-th dimension of the policy vector $\pi^a$. The computation for $\text{PathIG}$ is shown in Equ.(1). We compute each agent's contribution $c$ to the state transition from $s_{t_1}$ to $s_{t_2}$ and analyze the agent with higher $c$ first. We further study the adjustment frequency and its effectiveness in Section 4.2

## 4 Experiments and Analysis

### 4.1 Experimental Setup

We consider a challenging set of cooperative StarCraft II maps from the SMAC benchmark [23] classified as *Easy*, *Hard*, and *Super Hard* scenarios according to the baseline algorithms' performance. The inherent differences among various methods and their training procedure (e.g., on/off-policy learning for value-based/policy-based methods) bring difficulties when comparing methods in a reasonably fair manner without introducing additional components (e.g., importance sampling [13, 33] for off-policy methods). To attribute any poor performance of policy-based methods to potential algorithmic limitations or poor training conditions (in particular, high variance due to small batch sizes or insufficient gradient steps), we follow [5, 39], training all methods with 32 parallel runners to generate trajectories and using batches of 32 episodes. We evaluate each method every 320K steps with 32 episodes and report the 1st, median, and 3rd quartile win rates across 5 random seeds. Detailed information about the scenarios and the experimental setup is shown in the Appendix.

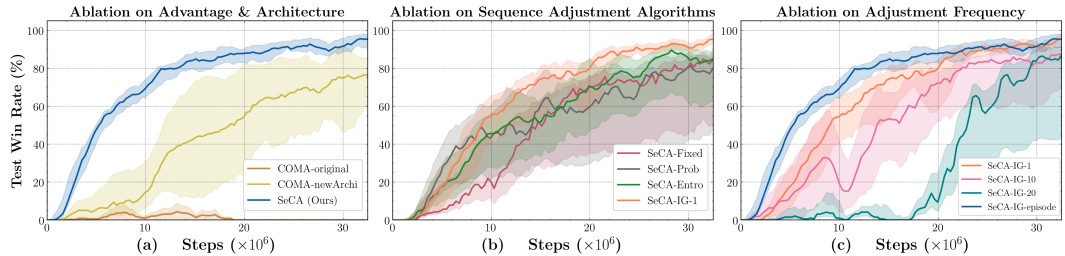

Figure 4: Ablations for SeCA's key elements on scenario `MMM2` (*Super Hard*). (a) investigates the effects of our sequential advantage and network architecture. (b) validates our sequence adjustment through integrated gradients. (c) shows the test win percentage with various adjustment frequencies.

## 4.2 Ablation Studies

We first carry out ablation experiments on a *Super Hard* map `MMM2` to validate key elements of SeCA.

**Proposed Advantage and Architecture.** In Section 3.3, we compare our sequential advantage with COMA's in two simple multi-agent particle environments and show our superiority in Figure 2. Afterward, we introduce a $f_\phi$ approximation and a corresponding network architecture. Here we apply the same approximation and architecture for COMA's counterfactual advantage (COMA-newArchi) and compare it with the original COMA and our method SeCA to show the effects of our advantage function, approximation, and network architecture. The result is illustrated in Figure 4(a). COMA performs poorly on this Super Hard map but acquires significant improvement with our approximation and architecture. Our sequential advantage further accelerates and stabilizes the training.

**Sequence Adjustment Algorithm.** SeCA's credit assignment sequence is dynamic. We compare our method with some intuitive adjustments to validate its effects. One could first evaluate agents with higher current-action probability (SeCA-Prob) or lower policy entropy (SeCA-Entro), as these agents are more confident in their acts, and we can assess other agents based on their behaviors. Since SeCA-Prob and Entro get a new order at each step, to be fair, we set the path length in Equ.(14) to one, i.e., consider agents' contributions based on the transition from $s_t$ to $s_{t+1}$ (SeCA-IG-1). Figure 4(b) illustrates that SeCA-Prob and Entro learn better than the fixed method (SeCA-Fixed), but Prob has a larger variance than Entro. Fixed is better than expected, which we believe is because that the fixed sequence acquires adequate training. Our integrated-gradients-adjustment performs the best in win rates and stability, and the others have inferior performance and incredibly high variance.

**Sequence Adjustment Frequency.** We next consider how the sequence adjustment frequency in SeCA-IG affects the performance. Except per step adjustment (i.e., SeCA-IG-1), one could also update the sequence after a stage or an episode. If we change the credit assignment order for every episode during training (SeCA-IG-episode), then $\tau_{t_1}^{t_2}$ in Equ.(14) represents a whole episode. As for stage adjustment, it is hard to define a stage in these tasks, and the stage length varies in diverse maps. Here we set stage length to 10 and 20, respectively denoted as SeCA-IG-10 and SeCA-IG-20. As the results in Figure 4(c) show, IG-1 and IG-episode have similar final win rates. However, IG-episode converges more quickly with smaller variance. The reason for IG-10(20)'s mediocre performance and high variance may be because the stage length needs to be dynamically adjusted. Inappropriate adjustment frequency fails to adapt to the stage changes in the task and causes insufficient training for each sequence. We utilized SeCA-IG-episode in other experiments and will investigate dynamic stage learning in the future to improve stage adjustment.

## 4.3 Comparisons with State-of-the-arts

We compare SeCA with some competitive algorithms, including the representative explicit credit assignment method COMA, the policy-based implicit method LICA, the common-used baseline QMIX and QTRAN. Methods are evaluated on 6 scenarios, including 2 *Easy* ones (`2s3z`, `1c3s5z`), 2 *Hard* ones (`2c_vs_64zg`, `3s_vs_5z`), and 2 *Super Hard* ones (`MMM2`, `3s5z_vs_3s6z`). We train all methods for 32 million steps in *Easy* maps and 64 million steps in *Hard* and *Super Hard* maps. These scenarios involve homogeneous and heterogeneous teams, symmetric and asymmetric battles, allowing a holistic study on all methods. Our experiments are based on the latest PyMARL [23]

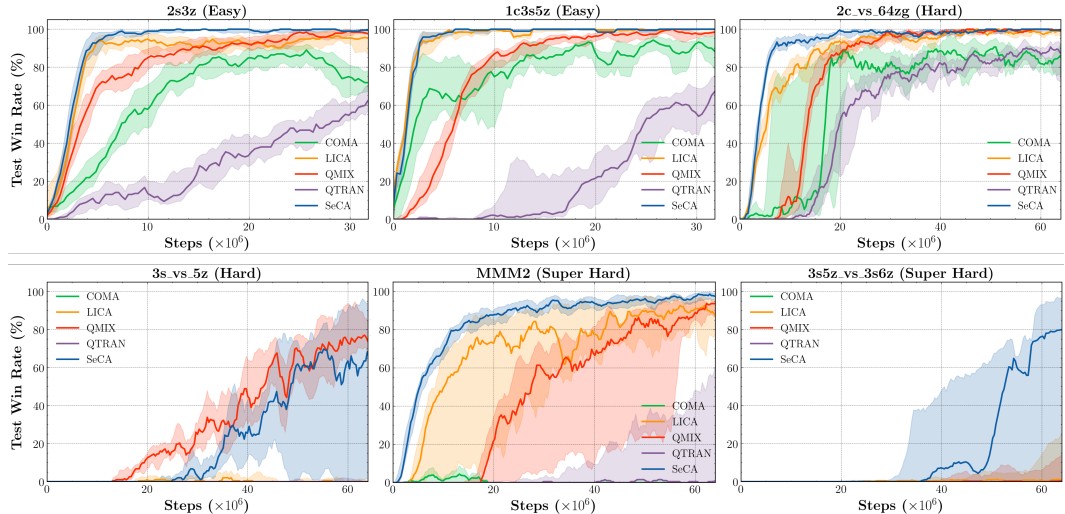

Figure 5: The comparison of SeCA against various baseline algorithms on six SMAC maps.

utilizing SC2.4.10. Performance is not always comparable between versions, so the results may be subtly different from the original papers.

As we can see in Figure 5, SeCA demonstrates its robustness by achieving good performances in scenarios with various characteristics. All methods except COMA and QTRAN solve two *Easy* scenarios, and SeCA performs better in convergence speed and stability. SeCA's advantage is further extended in the *Hard* map `2c_vs_64zg`, and it converges significantly faster than other methods. Although classified only as *Hard*, `3s_vs_5z` invalidates most algorithms except QMIX and SeCA, as Stalkers have to learn dispersing and making enemies give chase while maintaining enough distance ("kiting" technique) in this map. SeCA has a higher variance than QMIX. This is possibly because the Stalkers' scattering prioritizes individual performance over cooperation which is more in line with QMIX's monotonicity constraint. Nevertheless, SeCA's performance improvements on the *Super Hard* scenarios `MMM2` and `3s5z_vs_3s6z` demonstrate the effectiveness of our method. LICA's performance in `3s5z_vs_3s6z` here is different from the original paper, as the original results for this map are obtained by using a different entropy coefficient, which is explained in its open-source implementation.[1] This parameter tuning is unfair when comparing methods, so all experiments in this paper use the fixed entropy coefficient. We also visualize the learned sequences in different battles of `3s_vs_5z` to provide insights into our sequence adjustment in the Appendix.

We are supposed to compare our method with QPD that also utilizes integrated gradients to show our improvement. However, QPD modifies the original SMAC environment to acquire additional information for policy training, which is mentioned in its open-source implementation.[2] Therefore, it is unfair to compare QPD's learning curves in the modified environment with other methods, and QPD's authors did not provide methods' learning curves comparison in the original paper. We follow them, providing a win rate table in the Appendix to show our superiority over QPD.

## 5 Conclusions and Future Work

This paper presents SeCA, a cooperative MARL framework with sequential credit assignment. SeCA computes counterfactual advantage functions to evaluate each agent based on the actions of the preceding agents under a specific sequence. The sequence is adjusted dynamically according to agents' contributions to the team deduced by integrated gradients. SeCA accelerates policy convergence and improves the final performance over existing recognized methods in practice. In the future, we will further investigate stage learning in an episode and adjust the sequence per stage to improve SeCA and achieve adaptive cooperation in various task situations.

---

[1] `https://github.com/mzho7212/LICA`

[2] `https://github.com/QPD-NeurIPS2019/QPD`

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
