# OpenReview forum: "Cooperative Multi-Agent Reinforcement Learning with Sequential Credit Assignment"
_NeurIPS.cc/2021/Conference — NeurIPS 2021 Submitted_

### Official Review · Reviewer_c8rv · 2021-07-16

**Rating:** 6
**Confidence:** 3

**Summary:**

The authors are studying cooperative marl with global reward,  and in particular the credit assignment problem. This is considering the sequential approach where one assign credit one agent at a time by marginalizing out its action akin to difference rewards. When there is a natural sequence of importance of the agents and whose contribution depends on whose actions. They use CEO and staff as an illustrative example of when it makes sense.


**Limitations And Societal Impact:**

It is not clear to me if the sequential approach can be bad for all orderings in some situations, but I would guess that such exists.

**Main Review:**

The key novelty in this paper is that they adjusts the sequence based on a particular way of measuring the contribution of each agent , integrated gradients, that has previously been introduced for this purposes (decomposing global Q into ind agents) in QPD under the extra assumption of a local reward being linearly related to the contribution to the total. Here such local rewards and assumptions about them are not relied on and instead this value is used to rank the agents for sequential credit assignment.

The empirical evaluation on SCII micromanagement tasks looks promising.

The paper is combing a few existing techniques in a particular manner that feels slightly ad-hoc, though not without some intuition.

It would be interesting to know if the sequential credit assignment also reinforces the chosen ranking of contributions in some way and make the learning, in any way, such that the lower rank learns to "consider" the higher ranked more than the other way around.


**Time Spent Reviewing:**

3 hours

---

> ### Author Response · Authors · 2021-08-09
> **Authors' response**
>
> We thank the reviewer for the insightful review and comments! We addressed your questions and concerns below and revised our paper in line with your suggestions.
>
> **1. The novelty and originality of our work**
>
> We appreciate your comments about the novelty of our work, and here are some additional notes to help you make a more accurate assessment of SeCA's novelty. As you recognized, "adjust the sequence based on a particular way of measuring the contribution of each agent" is indeed an innovative point of our work. However, it is not the only and main innovation. SeCA is the first method that attempts to sequentially evaluate agents in the credit assignment process to the best of our knowledge. The proposed sequential credit assignment has achieved excellent results and has been recognized by other reviewers, one of whom noted that "a successful framework for sequential credit assignment has been a missing element of prior work on multi-agent RL."
>
> SeCA is not "combing a few existing techniques". Among the three elements presented in Sections 3.2 to 3.4 (sequential model, sequential credit assignment, and sequence adjustment), only the integrated gradients method used for sequence adjustment is an existing method. RUDDER and QPD have utilized this method before us. Although QPD was the first one to leverage this method in MARL, it has a strong assumption. SeCA eliminates this assumption and achieves more desirable results, providing new insight into the use of integrated gradients. Although we do not propose a new approach to adjust the sequence, we did sufficient ablation studies to explain why we chose integrated gradients and provided completely new insights into its usage. These attempts meet the requirements for innovative papers on the NeurIPS homepage, and we believe they are valuable to the MARL community.
>
> The sequence adjustment is only one part of SeCA, and the way we use integrated gradients provides new insights for the community. We would be appreciative if you can reassess the novelty of the entire sequential credit assignment framework based on these two points.
>
>
>
> **2. Whether the lower-ranked agent trained by our method learns to "consider" the higher-ranked**
>
> Thank you for this thoughtful question.
>
> We visualized the learned sequences in different battles of *3s_vs_5z* (SMAC) to provide insights into our sequential credit assignment in "Section E Visualization" in the Appendix. These visualizations showed the rationality of the sequential credit assignment to a certain extent. However, SMAC has no typical scenario to answer your question "if the lower-ranked learns to consider the higher-ranked". Therefore, to answer your question, we visualized the illuminating multi-agent particle experiment “Cooperative Navigation” shown in Figure 2(b) of our paper. Demonstrative videos are available at https://imgur.com/VFnDjCG.mp4.
>
> As seen in SeCA's demonstrations, once two agents choose the same target, one of them will immediately change its target based on the other's decision. However, the COMA-trained lower-ranked picks the same target as the higher-ranked agent and collides with other agents. This set of comparisons clearly illustrates that the lower-ranked agent trained by our method learns to "consider" the higher ranked, thus making the multi-agent system well-organized and gaining more total rewards. These visualizations further demonstrate SeCA's effectiveness and superiority.
>
> We are not very sure the meaning of your question "if the sequential credit assignment also reinforces the chosen ranking of contributions in some way", but we hope the videos and our response above could answer this question. If they are not, we are happy to further discuss with you in the rolling discussion period.
>
>
>
> **3.  If the sequential approach can be bad for all orderings in some situations**
>
> Thank you for this helpful question, and we will organize a new limitation section to help readers get a more comprehensive view of our method in the revised paper. Although SeCA can deal with multi-agent systems containing complex reticulated cooperative relationships, we cannot guarantee that the sequential credit assignment framework will perform optimally in all cases. SeCA is the first method that brings insights into sequential credit assignment to the MARL community, and we will continue to improve it, such as leveraging ideas like coordination graphs and dynamic stage learning. The added section will introduce the above situation and the sequence adjustment by stages discussed in Section 5.

---

### Official Review · Reviewer_E7Ut · 2021-07-16

**Rating:** 6
**Confidence:** 4

**Summary:**

This work proposes a novel approach for addressing the challenge of credit assignment in the context of centralised training with decentralised execution (CTDE): sequential credit assignment (SeCA). The method proposes to calculate the counterfactual advantage for each agent in a sequential manner, as well as an approach based on integrated gradients to dynamically determine the order for evaluating the agents (i.e., in descending order of the agents' contributions to the state transition).

**Ethical Concerns:**

No ethical concerns.

**Limitations And Societal Impact:**

No potential negative societal impact.

**Main Review:**

Originality:
The work addresses an important problem in MARL, namely credit assignment. The work extends the idea of calculating counterfactual advantage values and also incorporates the integrated gradients method for a novel purpose.

Quality:
I do not agree with the presentation and some of the statements made in Section 3.2. At lines 137-138 the work introduces a random variable to model the event that the credit assignment study is precise. You then proceed to unfold the joint probability distribution according to the product rule, on the 3 agents example, which is fine. But in lines 153-155 it is stated that: "The sequential MARL framework reduces the complexity of the model with six dotted arrows that indicate correlations between agents’ evaluations in Figure 1(b) by half, as those three dotted lines in Figure 1(c)". This is incorrect and misleading. First of all, these are 'dependencies' and not 'correlations'. Secondly, Figure 1(c) is not correct. The top nodes are no longer just the variables O1, O2 and O3, but rather: O1, O2|O1 and O3|O2,O1, obtained from the sequential calculation of the terms in the product rule.

Clarity:
The paper is nicely written and organised. I find that perhaps the recurring CEO-Staff example takes too much space for the utility it brings to the paper. It might have been more interesting to present for example a short discussion on a testing episode (from the supplementary material). This would show on a real example how exactly the idea of agent contributions translates in practice.

Lines 178-179 state 'After assessing agent a, we fix its action and evaluate agents after it based on a’s fixed action', can you clarify how or to what do you fix the action of each agent during the sequential assessment.

Significance:
I appreciate the ablation studies and comparisons with SOTA approaches. The study of the sequence adjustment frequency was interesting. I was wondering how the 10 and 20 values were selected for the stage adjustment? The experimental results show that SeCA performs well, on par or even better than the other considered approaches.

----Post-rebuttal----
I thank the authors for the diligent clarifications. I have raised my score.

**Time Spent Reviewing:**

4

---

> ### Author Response · Authors · 2021-08-09
> **Authors' response**
>
> Thank you very much for the review and comments! We updated our paper in line with your helpful suggestions and addressed your questions below.
>
> **1. The misleading content in lines 153-155**
>
> Thank you very much for this constructive feedback. We have followed your suggestions, changing the word "correlations" to "dependencies" and modifying the corresponding figure to make our paper more rigorous and precise.
>
> **2. The recurring CEO-Staff example**
>
> We appreciate this suggestion and have revised our paper correspondingly. We believed that the CEO-Staff example could clearly explain multiple enhancements of our method (including the origin of Equation 7 and the integrated gradient) and therefore repeated it in different sections. Although these repetitive examples provide various intuitions, it does take up too much space. We heeded your suggestion to remove some repetitions and presented a short discussion on a testing episode from the supplementary material.
>
> **3. How the 10 and 20 values were selected for the stage adjustment?**
>
> Each scenario in SMAC has a maximum episode length ranging from 60 to 400. To ensure generality, we keep the stage length within 60, including 5, 10, 20, and 30. Since good policies tend to end battles in much fewer steps than the maximum episode length, we dropped 30 (because it might make an episode only one stage); and since choosing 5 is too frequent for scenarios with big maximum episode length, we finally chose 10 and 20. As mentioned in lines 287-288 and 331-332, these two choices may be a bit crude, but we will continue researching the dynamic stages in the future, while this paper mainly focuses on the sequential credit assignment framework.
>
> **4. How do we fix the action of each agent during the sequential assessment?**
>
> This question is essential to understand our method, and we hope our response can bring you a deeper understanding of SeCA. We propose a novel advantage function shown in Equation 6 to enhance the COMA's (Equation 5). Since the computational complexity of this advantage increases exponentially with the agent number, we introduce a sequential model to reasonably simplify the computation. Note that our proposed sequential approach is only for evaluation, i.e., all the agents have chosen a fixed action at timestep $t$, and we assess these actions in a serialized manner.
>
> Returning to the CEO-Staff example, in evaluating the Staff, if we follow Equation 6, we need to compute the expected Q-value over all actions of the CEO. However, in the sequential model, the CEO's chosen action should be considered when evaluating the Staff's action. Therefore, we no longer need to compute the expectation over all the actions of the CEO but calculate the Staff's advantage function only based on the action chosen by the CEO at that moment. Instead of calculating the expected Q-value over all the possible actions of the CEO, we directly "fix" the CEO's action.
>
> Similarly, in a multi-agent system with $n$ agents, when assessing agent $a_i$, we need to consider the chosen action of $a_1, a_2, ..., a_{i-1}$, and the expectation over the actions of the agents behind $a_i$. Therefore, when calculating Equation 6, the expectation over $a_1, a_2, ..., a_{i-1}$​ can be simplified to the Q-value with fixed actions chosen by them, deducing Equation 7. We call this expectation replacement with the chosen actions from Equation 6 to Equation 7: "fix action". SeCA significantly reduces the expectation computation compared with Equation 6, assessing each agent accurately and more efficiently.
>
> We hope that this response will provide you with a better understanding of SeCA and that you will reconsider your rating of our work. If you have new questions about SeCA, we are delighted to discuss further with you in the upcoming session.

---

### Official Review · Reviewer_8jFv · 2021-07-21

**Rating:** 6
**Confidence:** 5

**Summary:**

This paper focuses on the problem of multi-agent credit assignment for cooperative multi-agent reinforcement learning. The problem is framed in a leader-follower manner, so it sees the agent actions in a sequential manner. The sequence is updated dynamically using integrated gradients as a measure of agent's contribution to state transition. The paper claims that this allows them to achieve state-of-the-art results on many star-craft (SMAC) benchmark tasks.


**Limitations And Societal Impact:**

The paper doesn't really discuss its own limitations explicitly. Adding a section would be quite helpful. There might be [more stochastic] scenarios that the sequential ordering may not make sense and might require leveraging ideas like coordination graphs [A,B,C] instead.

[A] W Böhmer et al. "Deep Coordination Graphs" https://arxiv.org/abs/1910.00091
[B] S Li et al. "Deep Implicit Coordination Graphs for Multi-agent Reinforcement Learning" https://arxiv.org/abs/2006.11438
[C] N Naderializadeh et al. "Graph Convolutional Value Decomposition in Multi-Agent Reinforcement Learning" https://arxiv.org/abs/2010.04740


**Main Review:**

**Strengths**

The prior of a specific ordering in the form of leader-follower definitely makes sense as a way to simplify the problem of credit assignment. Using integrated gradients to figure out the ordering dynamically is an even nicer trick. One important detail about the results that is not clarified in the paper itself is that these state of the art results are at the cost of more than an order of magnitude more samples than existing works. But it's a good exploration of how much data might be required for a lot of these methods to converge. The paper is also very clearly written (although gets a little repetitive with Boss-Staff analogies) and exploring alternative ideas for credit assignment in multi-agent reinforcement learning is quite pertinent to making these more practical.

**Weaknesses**

As [1,2] shows that these many samples are not necessary to achieve the performance that is being achieved. For example, [1] achieves 80% win rate on "3s vs 5z" with only 2million samples, while this paper seems to require 20 times more samples to reach 40% win rate. But improved credit assignment should definitely show up as lower sample complexity, so I am worried if the differences between the algorithms are not all just because of sub-optimal hyperparameters and difference in credit assignment is immaterial (I understand that there might be version differences in SMAC, but the other works [1,2] seem recent enough that it's unlikely that it's just a matter of version difference. I am not sure if following refs[5,39] leads to the best comparisons against other algorithms). Other figures only do ablation studies or comparison against COMA. In my own experience with these algorithms, COMA tends to underperform compared to just using a centralized baseline on most tasks. Moreover the proposed approach inherits the problem from COMA of not being able to scale to teams with larger number of agents.

[1] Witt et al. "Is Independent Learning All You Need in the StarCraft Multi-Agent Challenge?" https://arxiv.org/abs/2011.09533

[2] Hu et al. "Rethinking the Implementation Tricks and Monotonicity Constraint in Cooperative Multi-Agent Reinforcement Learning" https://arxiv.org/abs/2102.03479

Edit: Thanks to the authors for their comments. I'm still not convinced of the comparisons with off-policy methods but I'm not changing my positive impression of the paper.

**Time Spent Reviewing:**

4

---

> ### Author Response · Authors · 2021-08-09
> **Authors' response**
>
> Thank you very much for the constructive comments! We will follow these helpful comments in our revised version. Following are our responses to your questions and concerns.
>
> **1. The concerns on following experimental setup of refs[5, 39] and more samples than existing works**
>
> These two concerns essentially arise from the way to compare on/off-policy methods on the same scale. We will first explain why we followed the experimental setup of refs[5, 39] and then explain the reason for more samples based on this setup.
>
> Your concern about "if following refs[5,39] leads to the best comparisons against other algorithms" is understandable and has been considered when we conducted the experiments. A fair and reasonable comparison between on-policy and off-policy algorithms has been a significant concern for the RL community so far. The inherent differences across the baseline methods and their training procedure (e.g., on/off-policy learning for policy/value-based methods) make it challenging to juxtapose them without introducing extra components (e.g., importance sampling for off-policy evaluation) that could alter the baseline performance[1]. To this end, some methods scale down both the batch size and the number of batch updates for on-policy methods in an effort to align their sample efficiency against off-policy methods with experience replay; while reasonable, this makes it hard to attribute any poor performance of on-policy methods to either the poor training conditions (particularly high variance from small batch sizes and insufficient gradient steps) or the underlying algorithmic limitations. We prefer to focus on the latter rather than the performance gains from special techniques. Therefore, we followed refs[5,39], where all methods use batches of 32 episodes generated with 32 parallel runners to align the number of both batch updates and environment steps. These two references are accepted papers by NeurIPS, and we believe that their way of comparing on/off-policy learning for policy/value-based methods is well accepted.
>
> This experimental setup explains why our experiments "use more samples" than other works (off-policy methods). Our total interaction number (the horizontal axis coordinate) is higher than that in some other papers because SeCA used 32 actors to collect data in parallel, and the interaction of each actor with the environment is counted. However, this does not mean that we fed more samples to the network to train our agents. Off-policy algorithms such as QMIX and the ones you pointed out utilize one actor and reuse samples leveraging experience replay and importance sampling techniques. Their training utilizes a batch of samples sampled from the experience pool. In contrast, our on-policy method SeCA and refs[5, 39] use a batch of throwaway samples with the same batch size. Therefore, the sample number used to train SeCA is the same as those off-policy methods. The horizontal axis coordinates in our experiments are 20 times bigger than other works because we utilize 32 actors to interact with the environment. We hope this response could help you gain a new understanding of SeCA's efficiency.
>
> [1] Meng Zhou, Ziyu Liu, Pengwei Sui, Yixuan Li, and Yuk Ying Chung. Learning implicit credit assignment for cooperative multi-agent reinforcement learning. In *Advances in Neural Information Processing Systems*, 2020. (ref[39] in our paper)
>
>
>
> **2. A little repetitive with the Boss-Staff analogies**
>
> We thank you for this suggestion and have revised the paper correspondingly. We believed that the CEO-Staff example could clearly explain multiple enhancements of our method (including the origin of Equation 7 and the integrated gradient) and therefore repeated it in different sections. Although these repetitive examples provide various intuitions, it does take up too much space. We heeded your suggestion to remove some repetitions and added more experimental analysis to illustrate SeCA's superiority.
>
>
>
> **3. Only do ablation studies or comparisons against COMA**
>
> Our ablation studies compared only with COMA are not avoiding comparison with other methods or because of COMA's poor performances. The purpose of our ablation study is to present to the reviewers and readers why our method is effective and the improvement that each module brings. We only compared with COMA-original and COMA-newArchi because they could highlight the importance of our implementation and the proposed sequential credit assignment framework, respectively, as shown in Figure 4(a). Comparison with other methods in the ablation study would not show why SeCA is effective and would not visualize the magnitude of each module's effect. Therefore, our ablation study mainly focuses on COMA and COMA variants (the ablation study presented in Figure 4(a) also answers your following concern to some extent), and the comparisons with other baseline algorithms are shown in Section 4.3. Although the original COMA performs poorly, SeCA, which is also a policy-based explicit credit assignment method, achieves excellent results with our several improvements, outperforming well-recognized implicit credit assignment methods represented by QMIX and LICA, as shown in Figure 5.
>
>
>
> **4. Inherit the problem from COMA of not being able to scale to teams with a larger number of agents**
>
> This is a thoughtful concern. SeCA takes some inspiration from COMA, but it improves the key ideas and implementation. COMA does not perform well with many agents, and we think this is mainly because the computation of the counterfactual baseline is imperfect (see Lines 62-68 and 166-173 for details). When the agent number increases, the calculation of the advantage function for policy learning is increasingly biased, directly leading to COMA's poor performance.
>
> COMA is ineffective because of the unreasonable credit assignment. SeCA, however, focuses on this problem. We propose the sequential credit assignment, optimizing the computational complexity and improving the accuracy (see lines 174-186 and Figure 2) and correctness (see Claim 1 in line 195 for details) of the advantage function computation. In this way, when the agent number increases, SeCA still calculates the advantage function of each agent quickly and accurately to guide policy learning. Multiple experimental comparisons in the paper illustrate our improvements to COMA. The experiments in Section 4.3 also reflect that SeCA does not inherit COMA's shortcomings in scenarios with a larger number of agents (e.g., MMM2). We hope that this response will provide you with a better understanding of SeCA and that you will reconsider your rating of our work.
>
>
>
> **5. Adding a section about limitations**
>
> We thank the reviewer for this constructive suggestion, and we will discuss SeCA's limitations after Section 4.3 in revision.
>
> We attempted to illustrate that even agent systems with complex web-like coordination can utilize the sequential credit assignment framework in Lines 221-232 and 141-152. Note that in our method, the cooperative relationship and the evaluation model are two different things. There are still complex web-like collaborative relationships between the agents; we just carry out the evaluation sequentially. The sequential credit assignment framework is not restricted to simple linear relationships and can theoretically deal with common systems with web-like coordination. We hope that this explanation will help you gain a deeper understanding of SeCA's superiority.
>
> We believe that SeCA has potential defects, and we will take your suggestion to add a limitation section. Although SeCA can deal with multi-agent systems with complex reticulated cooperative relationships, we cannot guarantee that the sequential credit assignment framework will perform optimally in all cases. It may perform better if leveraging ideas like ref[A, B, C] you introduced in practice. In addition, we studied the influence of the sequence adjustment frequency in our paper, and the dynamic stage learning for the sequence adjustment we mentioned in Section 5 is still our research interest. We appreciate this valuable suggestion and will organize all these points into the new limitation section.
>
> **If you have further questions or concerns after reading our response, we are happy to discuss them with you through new comments.**

---

> ### Author Response · Authors · 2021-09-01
> **Authors' response 2 to the unresolved concern**
>
> Thank you very much for your positive impression of our paper! Here we further address your unresolved concern in the **following two aspects**:
>
> (1) A fair contrast between on/off-policy methods is still a significant concern in Single-Agent RL and Multi-Agent RL communities. A seemingly fair comparison solution would be to restrict the on-policy methods to sample the same amount of data as off-policy methods, but this would inevitably result in the on-policy methods using a very small batch size or reduce the number of iterations, which makes it challenging to analyze the performance of the on-policy methods. Its poor performance may be due to the batch and iteration limitation or the underlying algorithmic limitations. Considering the above problem, refs[5, 39] adopt a compromise solution to let both on-policy and off-policy methods use the same number of actors to interact with the environment to ensure that all methods utilize the same amount of samples. The starting point of this solution is to more accurately measure the performance of on-policy methods without having too much impact on the off-policy methods.
>
> Therefore, all methods utilized 32 actors to interact with the environment. Some papers (including RIIT you mentioned as ref[2]) choose to set the horizontal coordinates of the figures as "**Sampling steps per process**" when faced with such a problem. **Based on this experimental setting, the values of the horizontal coordinates of the experimental figures in our paper should be divided by 32 when compared with off-policy methods that use one actor in other papers. SeCA still maintains an outstanding performance under this comparison way.**
>
> **For example, as you mentioned, "QMIX in your ref[1] achieves 80% win rate on 3s_vs_5z with only 2 million samples, while QMIX in our paper seems to require 20 times more samples to reach 40% win rate." It happens because QMIX in our paper has 32 actors but has only 1 actor in ref[1]. QMIX in our paper achieves an 80% win rate on 3s_vs_5z with 64 million samples. It is the same when our horizontal coordinate is divided by 32. (64 million $\div$ 32 = 2 million)**.
>
> When writing our paper, we read refs[5, 39], which NeurIPS had accepted, and we considered that their experimental and graphical style should be well accepted. Therefore, we followed them, and the horizontal coordinate of "Sampling steps per process" was not used in our old version. **We have set the horizontal coordinate to "Sampling steps per process" (like RIIT) to facilitate a fair comparison** between off-policy methods using only one actor and on-policy methods in our modified version. We believe this is fairer for both on-policy and off-policy methods.
>
> (2) As for the win rate, **many methods, including IPPO (your ref[1]), utilize some tricks** in their experiments. For example, IPPO uses tricks including **changing network architecture, reward scaling, orthogonal initialization, layer scaling, observation normalization, eligibility traces, and careful tuning for each scenario**. These experimental tricks significantly improve the performance of IPPO. However, all the methods in our paper do not use these tricks. To fairly compare the underlying algorithmic performance, all the methods in our paper leverage the basic settings of the Pymarl framework. SeCA shows its high superiority over baseline methods in this widely used setting.
>
> We hope this response will give you a more comprehensive understanding of **SeCA's efficiency and effectiveness**. If it is still not convincing to you, we sincerely hope to hear from you and will be very happy to further discuss with you during the precious time before the end of the rolling discussion period. Thank you very much again for your suggestion and contribution to our work!

---

### Official Review · Reviewer_oyDk · 2021-08-03

**Rating:** 3
**Confidence:** 5

**Summary:**

Base on an observation that it is often difficult to determine the contribution a particular agent’s behavior has on the team when we have not well assessed other agents’ actions, this paper proposes a sequential credit assignment method (SeCA) which deduces each agent's contribution to the team's success one by one according to a particular order. Futher, according to this order, SeCA simplifies the computation of the advantage function of COMA (specifically, the expectation over other agents' actions) by fixing the preceding agents' actions. Finally, the sequence order is dynamically adjusted according to agents' contributions to the team deduced by integrated gradients.

**Ethics Review Area:**

["I don’t know"]

**Limitations And Societal Impact:**

Please see the main review above.

**Main Review:**


$\textbf{Pros:}$
(1) The motivation is very clear. The paper clearly argues that a successful framework for sequential credit assignment has been a missing element of prior work on multi-agent RL. I encourage the authors to continue pursuing this study as it should make a valuable contribution.

(2) The paper is well organized and clearly written.


$\textbf{Cons:}$

$\textbf{(1)} \text{ [Fatal error]}$ The implementation of SeCA has fatal errors. Specifically, as illustrated in Figure 3, when computing the expectation over agents' policies of the advantage function, i.e., $A^{a_{i}}=f_{\phi}\left(s ; u^{a_{1}}, u^{a_{2}}, \ldots, u^{a_{i}}, {\color{red}\pi^{a_{i+1}}, \ldots, \pi^{a_{n}}}\right)-f_{\phi}\left(s ; u^{a_{1}}, u^{a_{2}}, \ldots, u^{a_{i-1}}, {\color{red}\pi^{a_{i}}, \ldots, \pi^{a_{n}}}\right)$ (marked in red), the concrete implementation is absolutely wrong.

After checking the code provided in the Appendix, I found the author tries to compute the expectation over agents' policies by simply taking the probabilities of selecting all valid actions as input. Directly takeing the probabilities of selecting all valid actions (called policy vectors in the paper) as the input of the Critic neural network doesn't mean you're calculating the expected value!

This incorrect implementation will lead to wrong conclusion which will seriously affect the later studies.

I could understand that the author's purpose is to improve the efficiency of the expectation calculation. Correctness, however, must come first.

$\textbf{(2)}$ Based on the analysis in (1), the experimental evaluation results are not convincing.

$\textbf{(3)}$ [About the experimental results in SMAC] The experimental evaluation in SMAC is not significant. Most recently, the paper RIIT[1] shows that QMIX could consistantly achieve SOTA performance among all SMAC maps with some code level modification only, e.g., changing optimizer, incorporating TD($\lambda$) and enlarging exploration. So, I recommend the authors to re-implement SeCA based on pymarl-v2 (I call the optimized version of pymarl by Jian Hu in paper RIIT[1] as pymarl-v2) and show how it performs.

Overall, I vote for a rejection for current version of this paper. I may modify my rating if my concerns mentioned above are addressed.


$\textbf{Reference:}$
  * [1] Hu J, Jiang S, Harding S A, et al. RIIT: Rethinking the Importance of Implementation Tricks in Multi-Agent Reinforcement Learning[J]. arXiv preprint arXiv:2102.03479, 2021.
  * [2] Christianos F, Schäfer L, Albrecht S V. Shared experience actor-critic for multi-agent reinforcement learning[J]. arXiv preprint arXiv:2006.07169, 2020.

**Time Spent Reviewing:**

8 hours

---

> ### Author Response · Authors · 2021-08-09
> **Authors' response**
>
> Thank you very much for your detailed review and helpful comments! We will revise our paper in line with your suggestions. Following are our responses to your concerns.
>
> **1. Approximation of the expected value**
>
> We would first thank you for reading our paper and code with meticulousness and rigor. We would be appreciative if you can re-examine our implementation based on the following response and regard it as an approximation of the expected Q-value after careful consideration rather than an error.
>
> The precise computation of an agent's advantage function is shown in Equation 7 in the paper. It contains terms computing the expected Q-values over some agents' policies. The computational complexity of this calculation and the spatial complexity of specific implementations grow exponentially with the agent number. The accurate computation for the majority of scenarios in SMAC (teams with more than 3 agents) is unacceptable for the common computational resources at the time/space level. Therefore, approximations within a reasonable range are necessary.
>
> Our implementation was determined after careful consideration and initial experimental verification rather than an error. Our critic network is trained to estimate the Q-value when the input is the action vector $u=[u_1, u_2, ... , u_n]$, where $u_i$ is the one-hot representation of agent $a_i$'s action. We approximate the expected Q-value over $u_i$ by replacing $u_i$ with agent $a_i$'s policy vector $\pi_i$ as the input to the critic network.
>
> First, we illustrate in https://imgur.com/gV6aBWL.png with a straightforward but generalizable example that our implementation achieves a relatively accurate approximation of the actual expectation in the linear case.
>
> Inspired by the illustration above, we attempt to extend this approximation to the cases with activation functions. Although this conclusion does not hold in the more common nonlinear cases, we appreciate the simplicity and computational efficiency that such approximation brings. Therefore, we conducted experiments to verify the practical effectiveness of our approximation in nonlinear cases (the SeCA Architecture with ReLU) compared with calculating the actual expectation. The only difference between these two implementations is the way to acquire the expectation (our approximation vs. compute expectation).
>
> Since computing the actual expectation requires traversal of all action combinations, the vast intermediate data storage is unacceptable for our NVIDIA Titan V GPU, even without considering the time consumed. Therefore, we conducted experiments in SMAC scenarios with a small number of agents (*2s_vs_1sc, 3m, and 3s_vs_3z*). We only adjusted total training steps (3500000 or 5000000 steps), test interval (25000 steps), and parallel runners number (2 actors) for both implementations. The other experimental settings are the same as those in our paper.
>
> We attached the experimental results in https://imgur.com/xqMuHU2.png. Our implementation (SeCA-ourApproximation) has almost the same effect as calculating the actual expectation (SeCA-computeExpectation), proving the effectiveness of our approximation in practice. In fact, the ReLU activation functions we utilized "is more linear than other models like networks based on sigmoidal units"[1]. Therefore, our approximation is very close to the actual calculated value. It is worth mentioning that this approximation also dramatically reduces the time/space complexity and substantially improves the efficiency of the expectation calculation.
>
> We utilize this implementation instead of the actual calculation based on the theoretical guarantee in the linear case and the practical validation. Due to space limitations, we did not include our full thoughts in the paper, and we are sorry for burdening your understanding. We added related content in the revised version to facilitate a better understanding of SeCA. We also hope that you will re-evaluate our implementation in light of our above response. If you have further questions about this implementation, we would be happy to discuss them with you in further detail during the discussion session.
>
> [1] Goodfellow I, Bengio Y, Courville A. Deep learning[M]. MIT Press, 2016.
>
>
>
> **2. Further improvement on SMAC experiments based on pymarl-v2**
>
> We thank you for this constructive feedback. We have adopted your suggestion, and the supplementary experiments have been carried out.
>
> In completing this paper, we noticed the excellent results achieved by RIIT. However, our primary goal was not to achieve SOTA but to show the superiority of SeCA (i.e., the sequential credit assignment framework, which is still a missing element in MARL as recognized by you). Based on this consideration, we adopted the more widely accepted pymarl-v1 framework and its parameters to compare with other well-recognized methods that also built on pymarl-v1. This fair comparison highlights the superiority of our approach. After reading your suggestion, we agreed that the performance is also essential, so we immediately supplemented the experiment based on pymarl-v2. After our initial adjustments with some code level modification, SeCA's performance has been further improved. We sincerely appreciate this valuable suggestion. We will add the new results of implementation based on pymarl-v2 to the revised version and add these two helpful references you mentioned to our paper.

---

> ### Author Response · Authors · 2021-08-30
> **Authors' response 2**
>
> We want to express our deep gratitude for your detailed review and constructive comments. We provided explanations and answers to your questions through a new comment during the author response period, and we hope our response could address your concerns. If some part of our work is still confusing to you, we sincerely hope to hear from you and will be very happy to further discuss with you during the precious time we have left before the end of the rolling discussion period. We apologize for any burden this may cause and thank you very much again for your contribution to our work.

---

### Decision · Program_Chairs · 2021-09-27

**Decision:**

Reject

**Comment:**

The reviewers and AC discussed the paper. There was agreement that the approach is promising for dealing with sequential credit assignment in MARL. Nevertheless, there are concerns about the fairness of the experimental results and some of the technical details (e.g., the advantage function). It is great that the authors will do (or have already done) some additional experiments, but those are not currently visible. These results are necessary to understand the performance of the approach in the context of other state-of-the-art methods. Similarly, while the authors provide additional details about the approximation of the advantage function, this approach should be better motivated and clarified in the paper. In order to properly evaluate the paper, it needs significant updating and re-review.